# Development, Analysis, and Sensory Evaluation of Improved Bread Fortified with a Plant-Based Fermented Food Product

**DOI:** 10.3390/foods12152817

**Published:** 2023-07-25

**Authors:** Miriam Cabello-Olmo, Padmanaban G. Krishnan, Miriam Araña, Maria Oneca, Jesús V. Díaz, Miguel Barajas, Maristela Rovai

**Affiliations:** 1Biochemistry Area, Department of Health Science, Public University of Navarre, 31008 Pamplona, Spain; 2Dairy and Food Science Department, South Dakota State University, Brookings, SD 57007, USA; 3Pentabiol S.L., Polígono Noain-Esquiroz s/n, 31191 Pamplona, Spain

**Keywords:** bread, functional food, fermented, novel food, sensory, fortification

## Abstract

In response to the demand for healthier foods in the current market, this study aimed to develop a new bread product using a fermented food product (FFP), a plant-based product composed of soya flour, alfalfa meal, barley sprouts, and viable microorganisms that showed beneficial effects in previous studies. White bread products prepared with three different substitution levels (5, 10, and 15%) of FFP were evaluated for physical characteristics (loaf peak height, length, width), color indices (lightness, redness/greenness, yellowness/blueness), quality properties (loaf mass, volume, specific volume), protein content, crumb digital image analysis, and sensory characteristics. The results revealed that FFP significantly affected all studied parameters, and in most cases, there was a dose–response effect. FFP supplementation affected the nutritional profile and increased the protein content (*p* < 0.001). The sensory test indicated that consumer acceptance of the studied sensory attributes differed significantly between groups, and bread with high levels of FFP (10 and 15% FFP) was generally more poorly rated than the control (0%) and 5% FFP for most of the variables studied. Despite this, all groups received acceptable scores (overall liking score ≥ 5) from consumers. The sensory analysis concluded that there is a possible niche in the market for these improved versions of bread products.

## 1. Introduction

Consumer behaviors have changed dramatically in recent decades, probably because of increased concerns about the impact of food quality on overall health. This situation encourages healthy eating habits by reducing the consumption of unhealthy products and promoting the selection of more nutritious foods [1]. Among these, functional foods (FF) have gained increasing acceptance in recent years worldwide [2]. They refer to food products that confer a beneficial effect beyond basic nutrition, and when consumed regularly, can promote health and protect against disease, even though they are not medication [3,4]. They render an invaluable health benefit to the consumer, driven mainly by a complex of biologically active components, including probiotic and prebiotic compounds, bioactive peptides, metabolites, and antioxidant molecules [5,6]. To illustrate, FFs have been proven to be helpful in many noncommunicable diseases, such as type 2 diabetes, cardiovascular disease, and cancer [4], as well as in improving intestinal barrier function [7] and in alleviating food intolerance [8].

Due to food producers’ increasing interest in healthier food products, considerable efforts have been made to develop improved foods. A promising approach involves enhancing the nutritional quality of the marketed products. Bread, traditionally made of wheat flour, is one of the oldest foods and is widely consumed in most cultures [9,10]. It is an important part of the Mediterranean diet and can take part in a healthy diet [11]; however, regular bread often presents nutritional deficiencies (i.e., micronutrients, fiber, antioxidants) [12]. There are some factors affecting the nutritional value of bread products, including the type of starter culture (sourdough vs. yeast fermentation) [13,14], degree of refinement (i.e., whole-grain flour vs. refined flour) [15,16], and type of fermentation (backslopping vs. Chorleywood breadmaking process) [17,18].

In the last few years, multiple investigations have been conducted to improve bread and bakery products’ quality. Several strategies can be implemented, such as decreasing salt [19,20] or carbohydrate content [21], or increasing protein content [22,23] in bakery products. Other strategies rely on formulating new bakery products by partially replacing wheat flour with alternative flours of greater nutritional value, commonly known as composite flours [10,24,25]. Some examples describe the enrichment of bread with composite flours made from cereals and pseudocereals (wheat, oat, spelt, rye, quinoa) [26,27], grains (amaranth) [24], fibers (fructans) [28], oilseeds (chia, sunflower, flaxseed, and pumpkin) [29], fruits and vegetables (blueberry and grapefruit) [30], tubers (tiger nuts and sweet potato) [31,32], and by-products from the food industry [33,34]. Few studies have attempted to include fermented ingredients in bread formulations, using yoghurt [35], kefir [36], and fermented quinoa flour [27]. Curiously, even traditional bakery products, such as Iranian barbari bread [37] or Chinese steamed bread [38], have been fortified with different ingredients that increase their nutritional properties.

Frequently, changes in food formulation can cause unexpected changes in the physicochemical and sensory properties of the final product [39,40], and substantial modifications in traditionally marketed products could be rejected by regular consumers [41]. For example, the incorporation of some ingredients can lead to significant changes in the final products’ odor [42] or color [43], and thus condition consumer choice. In this line, some authors have described food neophobia, the reluctance to try new and unfamiliar food, as strongly influencing consumers´ acceptance of new FF products and thus consumer choices [44,45].

The present study examined bread fortified with different concentrations (0, 5, 10, and 15%) of fermented food product (FFP). FFP primarily comprises legumes and cereals, and also includes live microorganisms and fermentable compounds, as well as bioactive ingredients generated during the production process, collectively making it a health-promoting product. FFP was initially intended for livestock supplementation, and previous studies have shown beneficial effects in animals, including improved health status in rabbits [46] and dairy calves [47], and a similar product by the same producers provoked a greater energy efficiency for milk production and fiber digestibility in lactating goats [48]. Additionally, preclinical studies in type 2 diabetes rats fed FFP showed that the product protected against the development and progression of type 2 diabetes [49], and two other studies with a probiotic bacteria isolated from FFP also showed beneficial effects in type II diabetes mellitus management [50,51].

In this context, we attempted to develop a novel food product from bread by using FFP as a high-nutritional-value component, to serve as an alternative vehicle for providing the product to humans. The objectives of the present study were (1) to develop bread with different enrichment levels of FFP; (2) to evaluate the influence of different FFP replacement levels on the physical characteristics, color indices, quality properties, protein content and crumb structure of the final bread products; and (3) to explore whether the incorporation of FFP in the studied proportions led to products with acceptable sensory properties.

## 2. Materials and Methods

### 2.1. Product Description

FFP is derived from a plant-based food product manufactured and distributed by a Spanish company (Pentabiol S.L, Noáin, Spain) and is commercialized for livestock supplementation (FDA Registered). Its main components are soya flour, alfalfa meal, and barley sprouts, along with a combination of lactic acid bacteria (LAB) and nonbitter beer yeast (marketed under industrial property protection) that goes through two fermentative processes. Its microbiological composition includes 2.0 × 10^5^ colony-forming units (CFU)/g total bacteria, 4.6 × 10^7^ CFU/g *Lactobacilli*, and 1 × 10^5^ CFU/g yeast and fungi. The nutritional and physicochemical characteristics of FFP are summarized in Appendix A. FFP harbors numerous microbial metabolites generated during industrial processing and partially preserves its microbial community during storage [52]. Thus, it can be defined as fermented food with hitherto undefined microbial content [53].

### 2.2. Experimental Design

We studied three bread formulations with different replacement levels of bread flour by FFP (5, 10, and 15% of bread flour substitution equivalent to Bread-5, Bread-10, and Bread-15 groups, respectively), and bread without FFP (Bread-0) was included as a control. The flour blends were prepared and tested in triplicate, for a total of 12 bread loaves. For the sensory analysis (See Section 2.9), 12 extra loaves were prepared (*n* = 4 each group).

The outcome measures included loaf dimensions (maximum height, length, and width), mass and volume, specific volume (SV), crumb and crust color, protein content in the bread products, crumb digital image analysis, and consumer panel evaluation.

### 2.3. Ingredients and Preparation

Bread wheat flour (Gold Medal Premium Quality, General Mills, Minneapolis, MN, USA), salt, sugar (pure granulated cane sugar, C&H, New York, NY, USA), active dry yeast (Red Star, Wisconsin, USA), and water were used in each experiment. Extravirgin olive oil (EVOO) (La Española Olive oil, Vilches, Jaén, Spain) was used as liquid shortening. All ingredients were purchased from a local grocery store and stored at room temperature (RT) until use. FFP provided by the manufacturer (Pentabiol S.L, Noáin, Spain) was obtained from the same production batch. FFP was processed as follows before carrying out the experiments: (1) it was milled to a fine flour using a 0.5 mm sieve; (2) the resulting flour was autoclaved at 121 °C for 15 min to sterilize the product and improve preservation; (3) flour was stored in a sterile glass container at 4 °C until use. Appendix A shows pictures of FFP at each stage and Appendix A compares the nutritional profiles of FFP and the bread flour.

### 2.4. Bread Production

A bread machine (SKG Automatic Bread Baker model 3920, Hong Kong, China) was used in the baking process to obtain a consistent baking method. This model included 19 automatic cooking programs, three loaf sizes (1, 1.5, and 2 lb), and three toast colors (light, medium, and dark).

Based on this experimental design, 12 bread loaves were prepared using different bread flour replacement levels. The ingredients and proportions of each blend are listed in Table 1. All liquid ingredients (water and EVOO) were weighed and carefully placed in a bread machine pan before dry ingredients (flour mixture, sugar, and salt) were incorporated. For Bread-5, Bread-10, and Bread-15 groups, corresponding amounts of FFP were added to the dry ingredients and adequately mixed. Activated dry yeast was then added to the mixture. Each mixture was cooked using the “Basic bread” cooking program selecting a 1.5 lb loaf and medium crust color. The baking program lasted 3 h and consisted of a 10 min kneading step, 3 min rest, 5 min kneading followed by 10 min rest, and a final kneading phase for 20 min. Subsequently, there were two rising phases of 42 and 40 min each. Finally, baking was performed at 100 °C for 50 min.

When the cooking program was completed, the loaves were removed from the pan and cooled at RT. The loaf mass and volume were then determined, and the loaves were packaged in Ziploc^®^ bags, labeled, and stored at 4 °C or −20 °C until further analysis.

### 2.5. Bread Physical Properties

The maximum height (to the top of the mound), length (to the greatest extent), and width of each bread loaf were measured using a regular ruler. The color attributes of bread were determined using a chromameter (Konika Minolta CR 414, Tokyo, Japan). For this purpose, the lightness (*L**), redness/greenness (*a**), and yellowness/blueness (*b**) of the bread crust and crumb were determined in triplicate for each bread loaf.

### 2.6. Protein Content

With the purpose to checking nutritional changes following bread fortification with FFP, protein content (%) was determined in each group according to the Dumas method with a Rapid N Max Exceed (Elementar, New York, NY, USA). Nitrogen (N) was determined following the AOAC Method of Analysis 935.36, and the protein content (%) in the sample was estimated as N × 6.25. Protein values were adjusted for dry matter.

### 2.7. Bread Quality Attributes

Once the loaves were cooked and cooled at RT, the mass, volume, and SV were determined. Loaf volume (mL) was determined using the mustard seed displacement method, according to the AACC 10-05.01 approved method. The SV was calculated by dividing the loaf volume by the load mass (cm^3^/g) as previously described [25,42].

### 2.8. Digital Analysis of Bread Crumb Structure

For deeper insight into the structural changes with different percentages of FFP, digital image analysis was carried out in all the groups using the C-Cell Image Analysis System (Calibre Control Intl. Ltd., Warrington, UK, version 2.0). Each bread loaf was cut transversally into 1 cm thick slices using a food slicer (Chef´s Choice 610, Avondale, PA, USA). The first two slices were discarded, and the third to fifth slices were used for analysis. To assess the differences between the cell structures of the different groups, we studied the following parameters: slice brightness, slice area (mm^2^), cells area (%), number of cells; main length (mm), cell volume (mm^3^), cell diameter (mm), and cell wall thickness (mm). Measurements were recorded in triplicate for each bread loaf.

### 2.9. Sensory Evaluation by Consumer Panelists

Sensorial analysis was performed to study consumer acceptance of experimental bread groups (Bread-0, Bread-5, Bread-10, and Bread-15). To evaluate the sensory attributes of bread, panelists and potential consumers at South Dakota State University (SDSU, Brookings, SD, US) were recruited via email. A total of 42 panelists (64% female) aged 20–54 from 16 different nationalities (American, Bangladeshi, Brazilian, Cameroonian, Chinese, Colombian, Ethiopian, Honduran, Indian, Iranian, Mexican, Native American, Nepalese, Sri Lankan, Turkish, and Venezuelan) were recruited from among the university students and staff. All participants were declared regular bread consumers. The subjects received written information about the test and all the participants provided their informed consent to participate in the study. The study was exempt from ethical committee review.

Twelve bread loaves were baked on the previous two days: four for the appearance evaluation and eight for the sensory evaluation. The samples for the appearance evaluation were frozen (−20 °C) and the samples for the sensory evaluation were refrigerated (4 °C) and kept in a plastic Ziploc^®^ bag for freshness. On the day of the sensory test, frozen and refrigerated samples were left at RT. Prior to the sensory test, the bread loaves were cut transversally into 1 cm slices. Slices were cut in half longitudinally, and the test sample consisted of one half-piece of each bread group, including both crumb and crust. The samples were coded with random three-digit codes, and the sequences were randomized. All groups were presented at the same time to the panelists, along with water and McIntosh apples to cleanse the palate between samples. The panelists were asked to evaluate the appearance, crumb and crust color, odor, hardness, taste, aftertaste, and overall liking of the codified samples using a questionnaire (Appendix A). For the evaluation, a 9-point hedonic scale was used, being 1 “terrible,” 5 “maybe good, maybe bad,” and 9 “great” (Appendix A). Values between 1–4 indicated that panelists rejected the product, whereas values equal to or greater than 5 indicated that panelists accepted the product. The questionnaire included an example of the scale to assist the volunteers. Liking of aftertaste was graded with either “yes” or “no”. Additionally, volunteers were asked about the most preferred group (consumer preference) and liking/rejection of whole-grain bakery products.

### 2.10. Processing of Data and Statistical Analysis

Each group (Bread-0, Bread-5, Bread-10, and Bread-15) was baked in triplicate, and each parameter was determined in triplicate, for a total of *n* = 9 in each group (except for loaf volume and mass, which were determined once). All data are expressed as mean ± standard deviation (SD). All statistical analyses were performed using SPSS software for Microsoft (IBM SPSS Statistics 20), and the significant level was established at α = 0.05. Analysis of variance (ANOVA) was conducted to test differences between treatments, and Tukey´s post hoc multiple comparisons were performed to determine where significant differences existed among the groups. Data from the sensory evaluation were analyzed using the Kruskal–Wallis test, and frequency analysis was performed using the chi-squared test. The Spearman correlation coefficient (ρ) was estimated to determine the linear association between some test variables with an interval scale, while Pearson correlation coefficient was used for variables with an ordinal scale. The results were interpreted according to the degree of association as very high (ρ = 0.9–1), high (ρ = 0.7–0.9), moderate (ρ = 0.5–0.7), or low (ρ = 0.2–0.5), and statistical significance was set at *p* < 0.05.

## 3. Results and Discussion

The results for each determination and the corresponding discussion are presented in the following sections.

### 3.1. Effect of FFP on Bread Physical Properties and Color Attributes

The effects of different FFP substitution levels on the physical properties and color attributes of bread were analyzed (Figure 1). Loaf peak height was negatively affected by FFP and significantly decreased at replacement levels greater than or equal to 10% (*p* < 0.001). A similar effect has been previously described for composite flours [10] or distillers’ dried grains with solubles (DDGS) [54], and it was associated with decreased swelling index and dilution of gluten protein, respectively.

With respect to loaf width, only Bread-5 differed from the control (Bread-0) (*p* < 0.05). Loaf length was unaffected by FFP, and no significant differences were observed between the groups.

Color indices were studied to determine the effect of FFP on the physical characteristics of the experimental pieces of bread and revealed that the treatments significantly influenced the crust and crumb color (Table 2). In the crust, the three color attributes decreased as the FFP level increased in the bread formula, indicating a darker (lower *L**), less red (lower *a**), and yellow (lower *b**) appearance with FFP inclusion (all *p* < 0.001). Compared with Bread-0, *L**, *a**, and *b** decreased 25.1, 24.1, and 35.8% in Bread-15, respectively. We also found a darker (lower *L**), and yellow (lower *b**) appearance in the bread crumb, with decreases of 30.9 and 2.8%, respectively, in Bread-15, compared with Bread-0 (*p* < 0.001 in both). The *a** parameter was profoundly altered by FPP and markedly increased with all the substitution levels (*p* < 0.001), especially in Bread-10, more than 3.5 times greater than Bread-0.

In agreement with the data presented above (Figure 1 and Table 2), Figure 2 (upper panel) shows that physical appearance differed significantly between the groups. The incorporation of FFP notably influenced bread loaves´ shape, height, and color. The natural color of FFP, which is darker than that of wheat flour, as well as the Maillard browning originating from the reaction between reducing sugars and amino acids during FFP sterilization [31,39,55] (See Appendix A), could explain the darker color found in Bread-5, Bread-10, and especially Bread-15, compared with Bread-0 (control). As the quantity of FFP in the bread formula increased, the loaves became denser and more compact, and these differences were visible in both cell structure and texture (Figure 2, lower panel).

### 3.2. Effect of FFP on Bread Quality Attributes

The FFP is a source of fermentable residues and postbiotic compounds, such as microbial compounds and metabolites [56,57], factors that can impact the structure of the dough and the bread and change the texture, physicochemical characteristics, and staling properties of bread products [28,58,59]. Considering this, the incorporation of FFP in bread formulation may alter bread rheology and quality attributes. As shown in Figure 3A, the mean mass of all the bread loaves was comparable (*p* > 0.05). This was expected because the bread formula was fixed to 1.5 lb loaves. However, in terms of volume, we found differences between groups with FFP levels equal to or greater than 10%, with 18 and 32% decreases in Bread-10 and Bread-15, respectively (*p* < 0.001) (Figure 3B).

SV is commonly used to define bread size [10] and quality [60], and is determined by gas retention capability, which is itself influenced by factors such as ingredients, particle size, and processing process [58,61]. As in the case of volume, SV decreased with greater amounts of FFP volume (*p* < 0.001) (Figure 3C), and such effect can also be perceived in the pictures in Figure 2. This result is in good agreement with the observations of Clark et al. [61] and Li et al. [38], who observed a negative correlation between the incorporation level of fiber-rich alternative ingredients and loaf volume, and attributed such effect to a lower swelling index due to changes in gas retention capacity. An exception was a study by See et al. that investigated the incorporation of pumpkin flour for bread formulation and found that the lowest fortification level (5%) provoked the greatest SV compared to the control formulation (0%) and other treatments (10 and 15%) [25].

The amount and type of protein in bread blench, which depends on the ingredients, also impact starch gelatinization and reduce swelling power [10,60,62]. This is in line with our results from protein content (See Section 3.3), which negatively correlated to SV (ρ = −0.84; *p* < 0.001).

Bread volume has been related to bread shelf life [59] and chewiness [38]. In response, strategies such as changing the presentation of the ingredients, for example, whole seeds vs. flour [29], or raw vs. popped grains [24], could be considered to minimize the impact of the treatments on bread volume.

On another note, previous studies investigating the compactness of bread products have reported that it can impact starch digestion and the glycemic response, insinuating that more compacted bread with reduced volume could led to a reduced glycemic impact [18]. FFP is also a good source of fiber, which also affects the glycemic response [63].

Considering the above, there is a potential for Bread-10 and Bread-15 to have a gentler glycemic response than control bread. Nevertheless, appropriate experiments are required to confirm this hypothesis.

### 3.3. Effect of FFP Replacement Level on Bread Protein Content

Regardless of its vegetal origin, FFP has a relatively high contribution of protein (44.5%). As expected, we found that the protein content in Bread-0 (9.4%) increased with the incorporation of FFP, reaching 10.4, 11.3 and 12.1% of protein content in Bread-5, Bread-10, and Bread-15, respectively. This indicated that even the lowest FFP supplementation significantly increased the protein content of the bread loaves (*p* < 0.001) (Figure 4). A similar trend was previously reported for Barbari bread supplemented with different levels of DDGS [34].

The protein content of the final product is influenced by the nutritional characteristics of the flour and other ingredients used during the bread-baking process. While the use of FFP (in this study), DDGS [34], and legumes (soy, fava beans) would have a positive effect on protein content, the incorporation of other foods less rich in protein, such as pumpkin flour [25] or some cereals [26], did not raise protein content in the final bread loaves. Nevertheless, other authors have pointed out that additional factors during baking, such as temperature, fermentation characteristics, or Maillard reactions, could also have a symbolic effect on bread protein content [37], and should therefore be considered.

According to the nutrition claims by the European Commission in Regulation (EC) No. 1924/2006, only food with at least 12 and 20% of the energy value by protein can be considered a “source of protein” and “high in protein”, respectively [64]. Considering that, only Bread-15 can be considered a source of protein. Previous authors, however, agreed that only bread types presenting 15–20% protein could be considered protein-rich [21]. The significant increase in protein content with FFP had little or meaningless effect on the nutritional characteristics of the final bread. Notwithstanding, considering the importance of the protein fraction on the glycemic food load and subsequent glycemic control [65], any strategy to increase the protein content in a food product is valuable and should not be despised. This is particularly important when the products contain white flour, which is a risk factor for type 2 diabetes and other noncommunicable diseases [66]. Indeed, incorporating 10 and 15% FFP led to a protein content similar to that observed in bread products elaborated with whole-grain flour [40].

### 3.4. Image Analysis of Experimental Breads

Image analysis data showed that FFP significantly affected the cellular structure of bread. A proportional decrease in slice brightness was found with increasing levels of FFP (Figure 5A; *p* < 0.001), which is in good agreement with data from color parameters, where FFP dramatically darkened crumb and crust color (Table 2 and Figure 2). The Spearman correlation coefficient confirmed a positive and significant high linear correlation between slide brightness and *L** value (ρ = 0.99; *p* < 0.001).

Both the cell area and the number of cells were only affected by FFP levels equal to or greater than 10% (Bread-10 and Bread-15), with a negative and positive effect, respectively (Figure 5B,C; *p* < 0.001). Indeed, we identified a very high negative correlation between the two parameters (ρ = 0.98; *p* < 0.001). In previous studies, however, incorporating novel ingredients, such as orange pomace [67] and distillers’ dried grains (DDG) [38] in bread and Chinese steamed bread, respectively, had the opposite effect and decreased the number of cells.

Cell volume increased in Bread-5 and Bread-10, while Bread-15 remained similar to the control (Bread-0) (Figure 5D; *p* < 0.001). The incorporation of FFP had a different effect on cell diameter and wall thickness, which were significantly reduced at replacement levels of 15% (Figure 5E,F; *p* < 0.01). No significant linear correlation was identified between cell volume and cell diameter (ρ = 0.40; *p* > 0.05) and wall thickness (ρ = 0.22; *p* > 0.05). A previous study investigated the impact of different replacement levels of breadfruit flour on bread quality parameters [61], and the authors also found a negative impact of this ingredient on slide area, cell diameter, and wall thickness. Contrary to our results, these treatments positively affected the cell volume.

Lastly, in disagreement with data from SV (Figure 3C), the main length and slice area were unaffected by any treatment (Appendix A). Similarly, we did not find a linear correlation between SV and main length (ρ = 0.004; *p* > 0.05). As shown in the digital photographs, there were also physical differences in the appearance of the bread slices within the groups. In addition to presenting a more irregular shape, bread with FFP (Figure 5H–J) had a more porous surface than the controls (Figure 5G). A similar effect has been described for bread enriched with DGG [34]. In addition, previous studies have shown a negative association between SV and bread loave density [68]. FFP is fermented and contains viable microorganisms (bacteria and yeasts) [52], and even though it was autoclaved, it is possible that some microorganisms survived the inactivation or formed spores. This could contribute to the formation of carbon dioxide during bread preparation, thus leading to a greater cell volume in the experimental bread loaves with FFP. On top of that, metabolites and products generated by microorganisms during baking can also significantly impact the physicochemical characteristics of bread and the rate of staling, potentially influencing consumer acceptance [59].

### 3.5. Consumer Sensory Study of Experimental Breads

Table 3 presents the mean data from the consumer panel evaluations and Figure 6 helps visualize the sensory study results. Overall, we found significant differences between the groups for all the sensory characteristics tested (all *p* < 0.001). The control (Bread-0) was liked significantly more in all the sensory parameters than the other experimental bread groups (Bread-5, Bread-10, and Bread-15), and obtained the best scores in general appearance and overall liking. Besides that, most scores were equal to or greater than 5 in all the parameters and groups, except for the general appearance in Bread-15 (4.1, “Just a little bad”). We can conclude that adding FFP to bread affected consumer evaluation, but consumers still accepted all products. When the mean scores were calculated for all parameters, the best punctuation (7.8, “Very good”) was registered for both crumb and crust color in Bread-0.

In Figure 6, we can see that the hedonic ratings of Bread-0 and Bread-5 were closer in most parameters, though they significantly differed. Notably, the impact of FFP on odor was adversely affected even at the lowest replacement level (5%) (*p* < 0.001). Despite this, all groups received punctuation closer to or greater than 5, which indicated that most panelists had a neutral or positive perception. This effect is likely due to the soya present in the FFP. Soya and other ingredients, such as chickpea [69] or sorghum [70], can significantly impact food taste, and in some cases, cause a bitter or beany taste that may compromise consumer acceptability mentioned [42,71]. Indeed, taste and aftertaste perceptions were also significantly affected by FFP incorporation (*p* < 0.001 in both). Organoleptic properties, particularly taste and olfaction, strongly influence consumer behavior and food acceptance [72]. Therefore, additional ingredients could be incorporated during the production process to mask undesired organoleptic attributes in bakery products with FFP to develop more appealing bread products attractive to consumers.

Bread color was significantly affected by FFP incorporation, as shown in Figure 2, Figure 5 and Figure 6. Accordingly, the panelists’ perception of crust and crumb color was also affected by FFP (*p* < 0.001), even at the lowest incorporation level (Bread-5). We believe this might have biased other organoleptic attributes and had an important effect on the perception of the sensory characteristics. Previous studies have indicated the impact of color on sensory evaluations, particularly in bakery products [9,43]. Darker bread is commonly associated with whole-grain bread varieties [40], which can influence consumer acceptance. In our study, however, when panelists were asked about their liking of whole-grain products, most (92.5%) confirmed that they like this category of bakery products.

On another note, other authors have concluded that pH, which is influenced by the fermentation process, significantly affects bread crumb color [61]. Although there was no information on the pH value of the experimental bread loaves in this study, it is plausible to find differences based on the composition of FFP.

The treatment also altered hardness perception, and its hedonic evaluation decreased with increasing FFP content (*p* < 0.001). This could be due to the higher percentage of humidity in bread loaves containing FFP. Unfortunately, the bread texture analysis was not performed.

At the end of the sensory test, panelists were asked to rate their group of choice, and most of them indicated Bread-0 (70.7%), clearly surpassing the other groups (*p* < 0.001). Curiously, Bread-10 (17.1%) was preferred over Bread-5 (9.8%), and only a small minority chose Bread-15 (2.4%).

Food neophobia can dramatically alter consumers´ acceptance of food. It can be affected by demographic, cultural, and social factors [44,73], as well as by individual factors (genetics, age, gender, and personality) [45,72]. Moreover, it has been described that the level of nutritional knowledge of the potential consumers is of great relevance as well [74]. Interestingly, clusters of consumers with similar sociodemographic profiles share food preferences. A previous study examined the different factors influencing the decision to purchase functional foods among a Polish population. The authors found important sex and age differences and concluded that women and older men have a greater interest in the product´s health properties and nutritional value than young men [75]. Similarly, the same authors reported that women and old men prefer cereal-based functional products, while young men are more attracted to meat-based products. According to the authors, a potential market niche for bread supplemented with FFP would be women, old men, and subjects with a university education. Nevertheless, we did not find any linear correlation between the preferred group and age (ρ = 0.17) and sex (ρ = 0.24) (all *p* > 0.05). Similarly, there was no significant correlation between the preferred sample and nationality (ρ = 0.10; *p* > 0.05); however, we found that the individuals who preferred Bread-10 and Bread-15 were from countries whose traditional gastronomy is full of flavored and tasty food (Brazil, Cameroon, Iran and Mexico). Therefore, it is likely that bread enriched with FFP would be better accepted in specific countries or cultures.

Health is an important motivation for consumers when choosing a functional food, and so does price [74]. Previous research indicates that consumers´ willingness to pay is impacted by their trust in functional foods, which in turn is affected by the food matrix [76]. Another study with Russian and German participants revealed an important cultural effect and country differences regarding consumer acceptance and food neophobia [73]. Alternatively, findings from one study on a Lithuanian population suggest that motivating factors other than health consciousness, such as social factors like conspicuous consumption, perceived self-control motivation, and susceptibility to descriptive normative influence, can also impact consumers´ preferences for FF [77]. Therefore, all the above-mentioned factors should be strategically studied and addressed to identify the target population of bakery products enriched with FFP.

### 3.6. Further Research

There is abundant space for further progress in analyzing the incorporation of FFP in bread products. More experiments should address key aspects such as FFP dosage, safety, and stability of the final product, and alternative delivery systems or additional ingredients should also be considered.

In the formulation, we used extra-virgin olive oil for shortening, which is the main source of dietary fat in the Mediterranean diet and is associated with many health benefits [78]. Because the shortening effects of oil are determined by the fatty acid composition of the fat source [79] and its oxidative stability [80], the use of olive oil may provide anti-retrogradation activity different from that of bread formulated with regular shortening. Therefore, it would be interesting to study the staling properties of bread with different levels of FFP and verify whether it alters bread predisposition for retrogradation and staling. If FFP incorporation would lead to fast staling, incorporation of antistaling agents into the formulation should be considered to preserve freshness.

On another note, FFP is a good source of fiber, and its incorporation as an ingredient can change the carbohydrate digestibility, sugar content, and glycemic index [81,82] in bread. Similarly, FFP is a plant-based product that is highly likely to contain antinutritional factors. Interactions between nutrients [83] and other aspects of industrial production such as fermentation and drying are known to influence protein digestibility [84]. These aspects can be further tested and predicted using different indices and scores, such as the predicted glycemic index [28], the contribution of digestible starch [81], and in vitro protein digestibility [27]. This is relevant because protein and carbohydrate digestibility can significantly impact microbiota composition [84,85,86], and carbohydrate and fiber assimilation can also significantly affect the glycemic response [28,63].

Regarding the sensory properties of the experimental pieces, they could be fully explored through a sensory analysis by trained panelists with sensory experience in evaluating different types of bakery products, including more texture and sensory scores [9]. It would help develop the most suitable recipe. Similarly, a texture profile analysis of the crumb using a texture analyzer like previous work [61,87] would help to evaluate the texture properties of the end-products.

Lastly, FFP is expected to provide many bioactive compounds in the baked products; however, the functional effects of bioactive components is definitely determined by their viability, which in turn is influenced by the matrix, food processing, storage, and digestion process [30]. For this, well-designed and controlled preclinical experiments and clinical trials are mandatory before making any health claim for novel food products with functional components.

## 4. Conclusions

This study is the first attempt to incorporate FFP into a food matrix to develop enriched food for human consumption. Our study aimed to create a bread product with improved nutritional properties that can generate interest in regular consumers of FF products. We evaluated three different substitution levels (5, 10, and 15%) of wheat flour for FFP and compared them with control (Bread-0). The quantity of FFP significantly affected all the studied parameters, including physical characteristics, color indices, quality attributes, protein content, and bread structure. The results showed that incorporating FFP into bread formulation could be a valuable strategy for increasing bread protein content. However, future studies should analyze the proximate composition of the final bakery products to determine whether there are also relevant differences in dietary fiber and other macronutrients such as fats and carbohydrates. Besides, the microbiological analysis of loaves with FFP could clarify whether spores or partially activated microorganisms interfere with dough fermentation.

In addition, FFP affected the sensory properties of bread, as indicated by panelists. Sensory tests revealed that the consumers perceived significant differences in the palatability and sensory attributes of the groups. Bread with different levels of FFP had worse sensory scores than the control in most of the studied characteristics, although consumers accepted all groups. On top of that, some panelists indicated their liking for some attributes in Bread-10 and Bread-15, suggesting a market niche for this type of product.

In conclusion, FFP could be included as a functional ingredient in bread or related products to decrease the use of wheat flour, thus potentially increasing the nutritional and functional properties of bread products. Whether the doses (5, 10, and 15% FFP) and the format of FFP (milled and autoclaved) used in this study could lead to significant health improvements in consumers remains unexplored, and could be evaluated in future trials.

## Figures and Tables

**Figure 1 foods-12-02817-f001:**
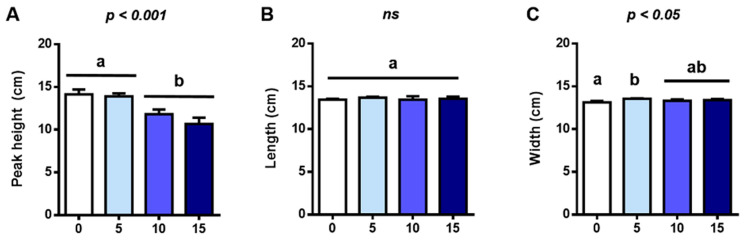
Effects of different replacement levels (%) of flour by FFP on the (**A**) peak height, (**B**) length, and (**C**) width of bread loaves. Different letters indicate significant differences (*p* < 0.05 and *p* < 0.001). FFP: Fermented food product.

**Figure 2 foods-12-02817-f002:**
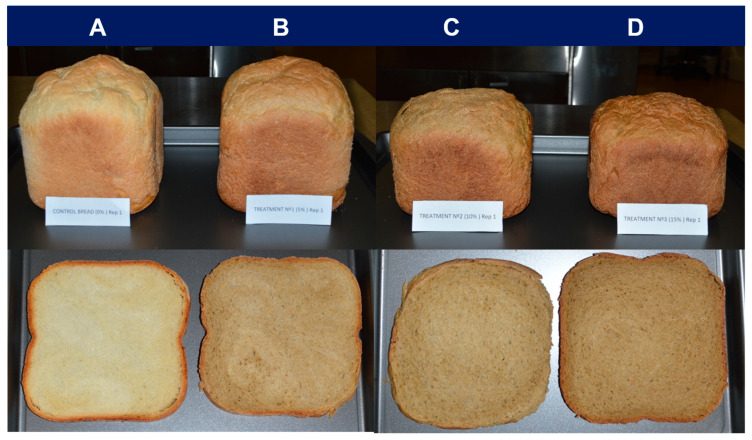
Photographs of bread loaves (**upper**) and central transversal slices (**lower**) of the final bread products baked with different replacement levels of FFP. (**A**): Bread-0, (**B**): Bread-5; (**C**): Bread-10, and (**D**): Bread-15. FFP: Fermented food product.

**Figure 3 foods-12-02817-f003:**
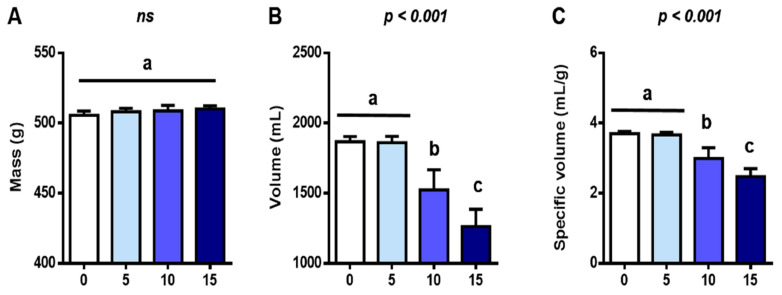
Impact of different replacement levels (%) of flour by FFP on bread (**A**) mass, (**B**) volume, and (**C**) specific volume. Different letters indicate significant differences (*p* < 0.001). FFP: Fermented food product.

**Figure 4 foods-12-02817-f004:**
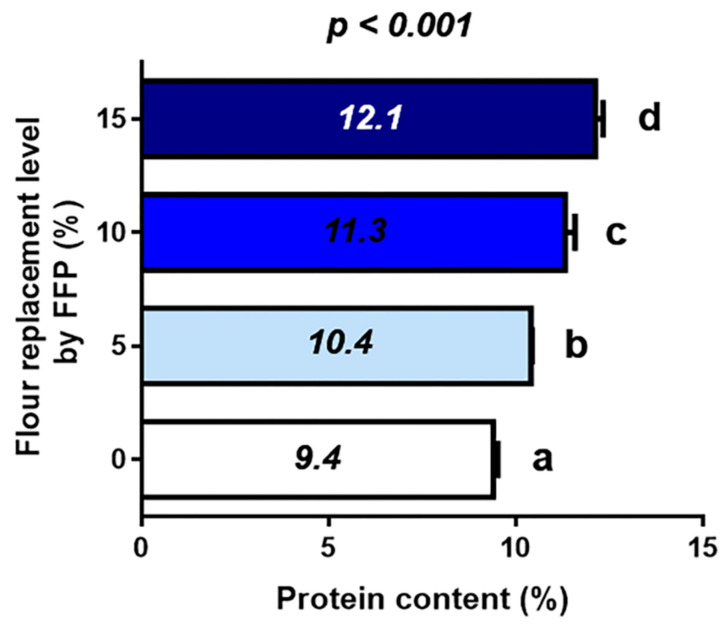
Impact of different replacement levels (%) of flour by FFP on bread protein content (%). Different letters indicate significant differences (*p* < 0.001). FFP: Fermented food product.

**Figure 5 foods-12-02817-f005:**
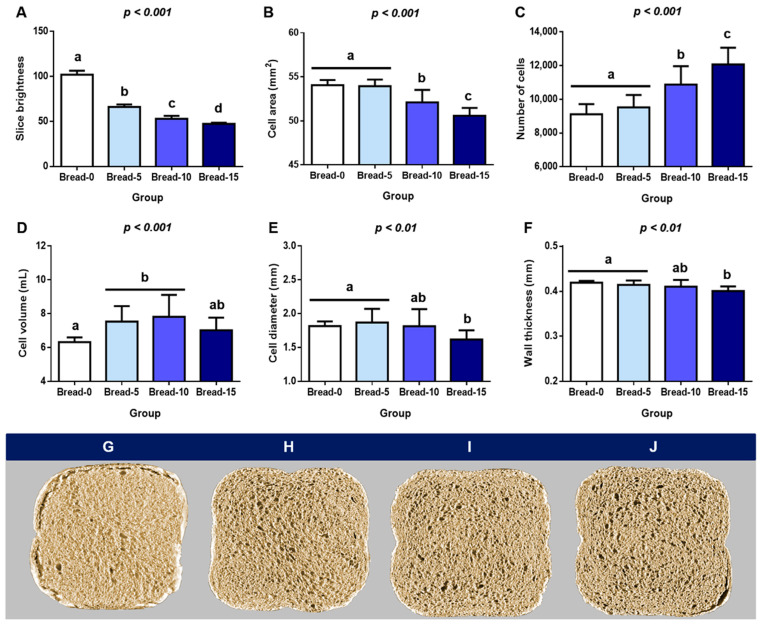
Effect of different replacement levels of FFP (0, 5, 10 and 15%) on (**A**–**F**) crumb properties determined by image analysis. High-resolution images of the bread slices with 0 (**G**), 5 (**H**), 10 (**I**) and 15 (**J**) % of FFP captured using the C-Cell software version 2.0. Different letters indicate significant differences (*p* < 0.01 and *p* < 0.001). FFP: Fermented food product.

**Figure 6 foods-12-02817-f006:**
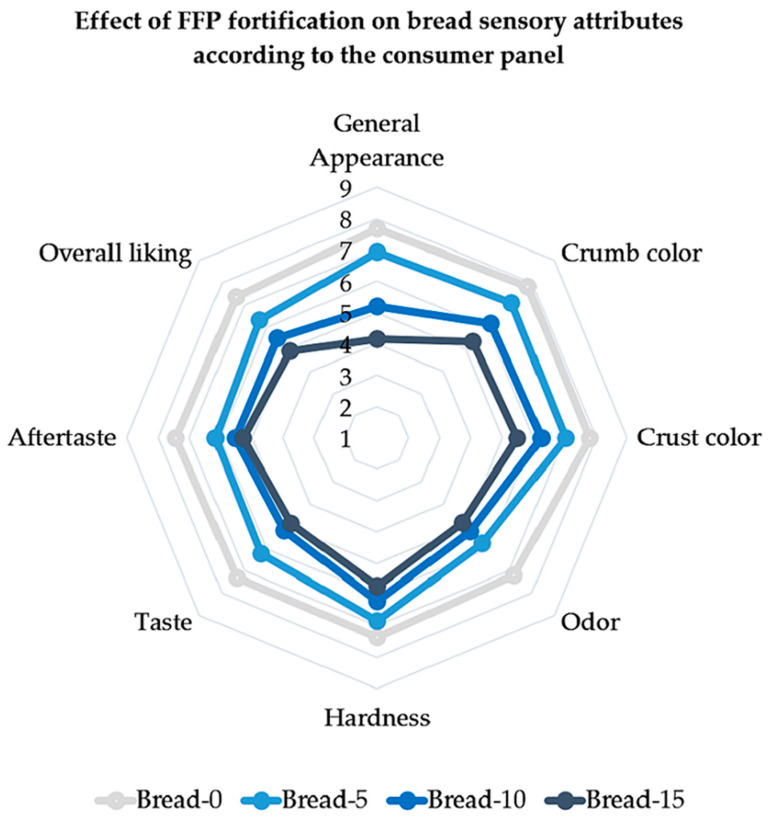
Effect of different replacement level with FFP (Bread-0, Bread-5, Bread-10, and Bread-15) on sensory attributes of bread loaves as evaluated using a 9-item hedonic scale. FFP: fermented food product.

**Table 1 foods-12-02817-t001:** Bread formulations.

Group	Ingredients (g)						
	Bread Flour	FFP	Sugar	Salt	Shortening	Activated Dry Yeast	Water
Bread-0	100	0	5	2	5	3	56.51
Bread-5	95	5	5	2	5	3	56.51
Bread-10	90	10	5	2	5	3	56.51
Bread-15	85	15	5	2	5	3	56.51

Ingredients based on a flour weight percentage. FFP: Fermented food product.

**Table 2 foods-12-02817-t002:** Effects of different FFP substitution levels on crust and crumb color parameters.

	Parameter	*L**			*a**			*b**		
	Group	Mean	SD	*p*-Value	Mean	SD	*p*-Value	Mean	SD	*p*-Value
Crust	Bread-0	52.5 a	5.3	*p* < 0.001	13.7 a	1.3	*p* < 0.001	27.9 a	3.3	*p* < 0.001
	Bread-5	46.1 b	2.4		12.0 b	0.4		23.8 b	1.6	
	Bread-10	42.8 bc	1.9		10.9 c	0.2		21.1 c	1.3	
	Bread-15	39.3 c	1.5		10.4 c	0.2		17.9 d	1.2	
Crumb	Bread-0	73.5 a	1.1	*p* < 0.001	-1.2 a	0.3	*p* < 0.001	21.8 a	0.7	*p* = 0.001
	Bread-5	58.8 b	1.3		2.2 b	0.3		21.9 ab	0.2	
	Bread-10	54.4 c	2.3		3.1 b	2.7		22.5 ab	1.3	
	Bread-15	50.8 d	1.2		2.9 b	0.4		21.2 c	0.4	

Means ± SD values in the same column followed by different letters are significantly different (*p* < 0.001). FFP: Fermented food product.

**Table 3 foods-12-02817-t003:** Means of sensory attributes of bread fortified with different levels of FFP.

	**Parameters**											
	**General Appearance**			**Crumb Color**			**Crust Color**			**Odor**		
**Group**	**Mean**	**SD**	***p*-Value**	**Mean**	**SD**	***p*-Value**	**Mean**	**SD**	***p*-Value**	**Mean**	**SD**	***p*-Value**
Bread-0	7.7 a	1.4	<0.001	7.8 a	1.0	<0.001	7.8 a	1.0	<0.001	7.2 a	1.3	<0.001
Bread-5	6.9 b	1.6		7.1 b	1.6		7.0 b	1.5		5.8 b	1.8	
Bread-10	5.2 b	1.6		6.1 b	1.4		6.3 bc	1.4		5.2 b	1.9	
Bread-15	4.1 c	1.8		5.3 c	1.6		5.5 c	1.6		4.8 b	1.9	
	**Hardness**			**Taste**			**Aftertaste**			**Overall Liking**		
**Group**	**Mean**	**SD**	***p*-Value**	**Mean**	**SD**	***p*-Value**	**Mean**	**SD**	***p*-Value**	**Mean**	**SD**	***p*-Value**
Bread-0	7.4 a	1.0	<0.001	7.3 a	1.2	<0.001	7.4 a	1.2	<0.001	7.4 a	1.1	<0.001
Bread-5	6.8 b	1.3		6.2 b	1.8		6.2 b	1.8		6.3 b	1.7	
Bread-10	6.2 bc	1.4		5.2 c	1.9		5.5 b	2.1		5.5 c	1.6	
Bread-15	5.7 c	1.7		4.9 c	2.1		5.3 b	2.0		4.9 c	1.8	

Means ± SD. Values in the same column followed by different letters are significantly different (*p* < 0.001). Scores on the 9-point hedonic scale were 1 = terrible; 2 = very bad; 3 = bad; 4 = just a little bad; 5 = maybe good, maybe bad; 6 = just a little good; 7 = good; 8 = very good; 9 = great. FFP: fermented food product.

## Data Availability

The data are available from the corresponding authors.

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
