# Peer review of "Development, Analysis, and Sensory Evaluation of Improved Bread Fortified with a Plant-Based Fermented Food Product"

_foods, 2023, doi:10.3390/foods12152817_

Round 1

Reviewer 1 Report

The aim of the manuscript, entitled “Development, analysis, and sensory evaluation of improved bread fortified with a plant-based fermented food product”, was the use of a fermented product for bread fortification, for this reason, the authors introduced the studies and the strategies used for bread fortification, and referred about healthy properties of fortified breads described in literature.

Despite the purpose of the paper, the experimental plan was developed to study other properties of bread, as physical and sensorial properties, and healthy outcomes were not reported at all.

This study deals more with food technology and food quality, but does not report any data regarding the nutritional and healthy properties of the bread fortified with FFP. Moreover, the results presented in this study does not show any novelty aspects.

The manuscript might be completely rewritten. Two possibilities:

1)      focusing on other properties of fortified bread, and not on the healthy properties. The introduction should report the effect of new ingredients, as the FFP, on the properties of bread, and the aim of the work must be changed. The scientific level of this paper would not be high.

2)      reporting data dealing with the nutritional and healthy properties of the experimental bread. This paper would be more interesting.

Below you will find some other comments.

I cannot refer to the line numbering in the manuscript, as it’s not present. In black find the sentences from the manuscript, in red the reviewer comments. 

Introduction

-          “In the last few years, multiple investigations have been conducted to improve bread and bakery products quality. Several strategies can be implemented such as ……………………. or developing gluten-free bread formulation”

I think gluten-free bread is not an improved bread, but it is a different bread, a different type of bread, which was “invented” for celiac consumers.  Remove the citation of gluten-free bread.

-          “and thus condition consumer elections”

May be you want to say “consumer selection” ?

-          “The present study examined bread fortified with ….”

-          “In this context, we attempted to develop a new FF from bread by using…”

The purpose of the work is quite confusing, and It’s not clear what kind of fermented food product (FFP) has been used to make this study. It is clear later on (reading M&M) that FFP is a plant-based food product manufactured and distributed by a company. The authors referred that “previous studies have shown beneficial effects in animals….” and they mentioned several studies that have been done using FFP, it seems that FFP used in this study is the same as the one reported in the bibliographic references [46-51], but it isn’t.

Reformulate the purpose of this study. It must be clear, indicating what has been used, what you wanted to obtain and how you performed it. 

Materials and Methods

2.8. Digital Analysis of Bread Crumb Structure

The authors should better describe the methods used to analyze bread porosity, i.e. which software and parameters used to measure pores. 

2.9. Sensory Evaluation by Consumer Panelists

Did you use frozen bread for sensorial analysis ? If yes, how did you thaw the bread (time and temperature) ?

Reviewer 2 Report

The authors of the work submitted for review have set the aims of the study 1) to develop bread with different enrichment levels of FFP, 2) to evaluate the influence of different FFP replacement levels on the physical, chemical, and rheological properties of the final bread, and 3) to explore whether the incorporation of FFP in the studied proportions led to products with acceptable sensory properties.

While objectives 1 and 3 were implemented in the research and achieved, objective 2 was treated very randomly. It is difficult to assess the effect of the FFP addition on the chemical composition of bread if only the protein content was determined. Other primary components should be investigated, e.g. fat, ash, carbohydrates. And if the authors consider that bread enriched with FFP is a functional product, this should be supported by the results, such as the content of bioactive substances, antioxidant activity. Also, the results regarding the rheological properties of the finished bread were not presented. In my opinion, the assessment of the crumb structure is not the assessment of rheological properties.

In chapter 2.10 lines 183-185 The authors describe that the obtained results were statistically processed using Spearman correlation coefficient and Pearson correlation coefficient. These results are not presented in the tables in the paper or in the supplement. They are only discussed in the text.

96- According to the authors, the breads were baked in a bread machine, i.e. in a mold of a certain length and width. So what was the point of this kind and analyzing these dimensions of bread, when the breadth and length of each baked dough was of the same kind? mid-to-height billet differences sprout up.

101 - no information what flour was used wheat or rye?

121 - Were the breads really baked at 100oC? Wheat bread is baked at a temperature of about 200-250oC. In my opinion, after 50 minutes at 100oC, 1.5lb loaves would not be baked.

142 - repetition of verses 125-126

152 - What's the point of analyzing the maximum height of a loaf and the maximum height of a slice cut from that bread? This is a duplication of the results.

180 - replace "at 0.05" with "at α=0.05"

197 - composite flours, DDGS expand these abbreviations

362 - the abbreviation DDG should be expanded

Table S1 The values of individual features in the table should be given with the same accuracy.

Reviewer 3 Report

Dear authors, after reviewing the article submitted to Foods entitled "Development, analysis, and sensory evaluation of improved bread fortified with a plant-based fermented food product", I am pleased to declare that the article was prepared and written with order and care. There are almost no errors detected and that speaks of the professionalism and commitment of the authors.

The summary is clear and concise, as well as the key words selected.

In the introduction, the aspects to be covered are touched upon in order for the reader to understand the reasons for the research.

Please in all references to temperature do not use the masculine ordinal indicator and use the correct degree symbol.

In the section on materials and methods, could the authors explain what were the reasons for autoclaving the FFP?

Line 155, sensory analysis section, it is not necessary to state that they were untrained panelists, it is understood when performing a sensory analysis of acceptance with a hedonic scale.

Line 157, what are the reasons for the authors to consider the panelists as potential consumers?

Line 158, why only 42 judges were used and not at least 50, as the authors defined the minimum number of judges needed for the trial?

The sections of analysis, results and conclusions are well written, cite tables and figures pertinently, for each phenomenon an explanation is given and adheres to the objective of the work. Only in line 386, the bread control was more... was more to the liking of?

Finally, according to the section on future research on the subject, it would be recommended to evaluate the real digestibility of the carbohydrates present in the product and of the proteins provided. In addition, the presence of olive oil, which mainly acts as a plasticizer to avoid retrogradation, would be prudent to be studied.

Round 2

Reviewer 1 Report

The authors have decided to maintain the setting of the manuscript, which is oriented on the healthy and nutritional aspects of fortified bread, in order do not “confuse” the readers, they say. I do not agree with their decision; I believe the readers can get confused, as they aspect to read results on the nutritional properties of bread, but results concern exclusively technological and sensorial aspects of bread. What the authors will consider to make in future works about the nutritional properties of this bread, is not important for this paper.

My point of view is the same; I don’t like how the manuscript has been structured, even after the revision and the improvements that were actually made to the text.       

 P.S. Have a look to the section 2.9.: the samples for the sensory evaluation were frozen or refrigerated ?

Author Response

Kind regards

Reviewer 2 Report

line 168 - expand the abbreviation RT

Author Response

Kind regards
